# Structural and Rheological Properties of Yanang Gum (*Tiliacora triandra*)

**DOI:** 10.3390/foods11142003

**Published:** 2022-07-06

**Authors:** Jittra Singthong, Ratchadaporn Oonsivilai

**Affiliations:** 1Department of Agro-Industry, Faculty of Agriculture, Ubon Ratchathani University, Warinchamrap, Ubon Ratchathani 34190, Thailand; jittrawara@gmail.com; 2Health and Wellness Research Group, School of Food Technology, Suranaree University of Technology, Nakhon Ratchasima 30000, Thailand

**Keywords:** Yanang gum, *Tiliacora triandra*, GC-MS, FT-IR, NMR, rheological properties

## Abstract

Plant polysaccharides are used in the food industry to improve the texture and stability of food. The viscosity of polysaccharides, which includes both thickening and gelling, is an important characteristic. Yanang, *Tilaicora triandra* (Colebr.) Diels., composed of polysaccharide gum in its leaves. In this research, Yanang gum’s structural and rheological properties were investigated. The gum’s structure is xylan, with a backbone made up mostly of mixed (1,3)- and (1,4)-D-xylan. The average molecular weight of Yanang gum is 3819 kDa, with a gyration radius of 120.4 nm and an intrinsic viscosity of 14.6 dL/g. The power-law model was found to be the best fit for Yanang gum flow curves. The consistency coefficient, k, increases significantly with concentration in both the forward and the reverse measurements, whereas the flow behavior index, n, decreased as concentration increased. Yanang gum exhibited shear-thinning flow behavior. Increasing the concentration results in heightened G′ and G″, and the cross-over point shifts toward lower frequencies. The results of this study show that Yanang gum may be beneficial as other natural gums for food products.

## 1. Introduction

Yanang, *Tiliacora triandra* (Colebr.) Diels., is in the family of *Menispermaceae*. It is widely distributed throughout mainland Southeast Asia [1]. Singthong et al. [1] reported the optimal extraction to prepare polysaccharide gum from Yanang leaves. The optimized extraction condition was shown at temperature of 85 °C, the ratio of leaf:water at 1:6.6 and extraction time for 100 min.

In addition, researchers have studied the major monosaccharide of Yanang gum and reported that it is composed of xylose together with natural sugars which when investigated by FT-IR spectra were similar to that of xylan. Moreover, Chaikham et al. [2] applied Yanang gum with maoluang juice to prepare mixed probiotics by spray drying process and then exposed it to in vitro gastrointestinal environments. The results showed that Yanang gum had an effect on the survival of *Lactobacillus casei* 01 and *Lactobacillus acidophilus* LA5 when compared with free cells, by increasing it around 15–20% and 20–21%, respectively. Furthermore, Yanang leaf extract has significant phenolic compounds and high antioxidant activities [3].

Plant polysaccharides are used in the food industry to improve the texture and stability of food. An important property of polysaccharides are their viscosity, which contribute to both thickening and gelling [4]. Yanang leaves are used to prepare a sour-tasting soup with bamboo shoots, chilies, salt, citric acid and other ingredients. The leaves are used to make the broth, primarily as a thickening agent. Rheological properties are important in understanding the structures of these compounds and the handling, processing, mastication and utilization of foods [5]. The rheological data of extrusion, pumping, mixing, agitation, heating, coating and process control have applications in process engineering analyses, quality control and shelf-life estimation, texture evaluation, product development and the development of constitutive equations for rheological characterization [6,7]. Functional properties of polysaccharides are mostly related to the physicochemical mechanisms underlying their behavior in an aqueous medium, and these mechanisms result from the thermodynamics of the system. Water solubility is related to solvent quality, and the strength of interactions between the polysaccharide and water. The hydrodynamic volume and the thickening properties are a consequence of water–polysaccharide interactions. However, gelling properties occur due to equilibrium between polymer–polymer and polymer–solvent interactions [8]. Thus, using polysaccharides, individually or in combination with others, it is possible to fabricate products using a knowledge of rheological properties. The active components in Yanang could be another source of natural polysaccharides. Singthong et al. [1] have recently shown that the polysaccharide gum from Yanang leaves is a xylan. However, the structure and molecular characterization of this xylan have not yet been determined.

The main objective of this study is to study the structural clarification and molecular description of xylan, including the steady and dynamic rheological properties of the polysaccharide gum extracted from Yanang leaves.

## 2. Materials and Methods

### 2.1. Sample Preparation

Yanang leaves, obtained from a farmers’ market in Northeast Thailand (Ubon Ratchathani province), were washed with water to remove dust and infected leaves and dried at 60 °C for three h. Further, the dried leaves were crushed and stored at room temperature in a vacuum-packed container before usage. After that, the Yanang polysaccharides were extracted according to a previously described method [1]. The extraction condition was at the leaf:water ratio of 1:6.6, 85 °C and 100 min extraction time. Then, the extract was filtered through a vacuum filter and centrifuged. The concentration process was carried out with the supernatant before being precipitated with ethanol. The residue was then dried in a vacuum oven, and Yanang gum was made by further grinding the residue.

### 2.2. Structural Characterization

#### 2.2.1. Chemical Composition

Monosaccharide compositions were analyzed using the method reported by Singthong et al. [9]. Yanang gum was hydrolyzed in 1 M H_2_SO_4_ for 2 h at temperature 100 °C to provide the constituent monosaccharides. The hydrolysate was cooled down, diluted and filtrated before being injected into an HPAEC column (Dionex HPAEC system composed of a pulsed amperometric detector (PAD), Dionex Canada Ltd., Oakville, ON, Canada).

#### 2.2.2. Linkage Analysis (Methylation and GC-MS of Partially Methylated Alditol Acetates (PMAA))

A methylation exploration was carried out according to Singthong et al. [10]. First, the dried samples were dissolved in anhydrous DMSO for 2 h at 85 °C. Also, the solution was constantly stirred and further sonicated for 2 h to ensure that the samples were completely dissolved. Next, the 20 mg of dried sodium hydroxide powder was added to the DMSO and stirred for 3 h at room temperature. Then, the methyl iodide (0.3 mL) was added and stirring continued for 2.5 h. Methylene chloride was used to extract partly methylated polysaccharides, which were then put down a sodium sulfate column to remove water and dried under a nitrogen stream. The samples were also hydrolyzed in 4 M trifluoroacetic acid (0.5 mL) in a sealed test tube at 100 °C for 6 h before being dried in a nitrogen stream. Before acetylation with acetic anhydride, the hydrolyzed acid was reduced with deuterated sodium borohydride (0.5 mL). The partly methylated alditol acetates (PMAA) in aliquots were then injected into a GC-MS apparatus (ThermoQuest Finnigan, San Diego, CA, USA) fitted with an SP-2330 column (Supelco, Bellefonte, PA, USA) (30 m × 0.25 mm, 0.2 μm film thickness, 160–210 °C at 2 °C/min, and then 210–240 °C at 5 °C/min) equipped with an ion trap MS detector.

#### 2.2.3. Spectroscopic Analysis FT-IR and NMR

All samples were dried and measured using the method reported by Singthong et al. [10]. The 128 scans were performed using a FTS 700 FT-IR spectrophotometer with a DTGS detector utilizing a Golden-gate Diamond single reflectance ATR in absorbance mode from 4000 to 400 cm^−1^ (mid-infrared range) at a resolution of 4 cm^−1^ (DIGI-LAB, Randolph, MA, USA). A Bruker AMX 500 FT spectrometer was used to record both ^1^H and ^13^C NMR spectra. All samples were dissolved in deuterium oxide (D_2_O) for 3 h at 90 °C before analysis. ^1^H, ^13^C, and 45° COSY (after pre-saturation) spectra were recorded.

### 2.3. Molecular Characterization

The molecular weight, molecular weight distribution, radius of gyration and intrinsic viscosity were all measured using high-performance size-exclusion chromatography (HPSEC) [4]. During the measurements, two columns were joined in series and kept at 40 °C (Shodex Ohpak KB-806 M, Showa Denko K.K. Tokyo, Japan; Ultrahydrogel linear, Waters, Milford, MA, USA). Viscotek Triple detectors (Viscotek Co., Houston, TX, USA), which featured a refractive index detector (Model 200), a viscometer (Model 250), and a right-angle light scattering detector, were utilized in conjunction with a Shimadzu SCL-10Avp pump (Model 600). The mobile phase was 100 mM NaNO_3_ containing 0.03% (*w*/*w*) NaN_3_ with a flow rate of 0.6 mL/min. The sample injection volume was 100 μL. The reference was pullulan, which had a known molecular weight and intrinsic viscosity, while the Yanang gum solution had a concentration of 0.146 mL/g.

### 2.4. Rheological Measurements

The rheological measurements were determined by ARES-controlled strain rheometer with Peltier temperature control (TA Instruments, New Castle, DE, USA). The sample dispersion was placed between parallel plates (50 mm diameter), and setting the gap between the two plates to 1 mm. 

#### 2.4.1. Preparation of Solutions

The Yanang gum solutions was prepared with concentrations varying from 0.1 to 3% (*w*/*v*) by dissolving the gum into appropriate quantities of distilled water, at 80 °C, whisking for 2 h before allowing it cool to room temperature.

#### 2.4.2. Steady Shear Measurements

The steady shear viscosity was measured at various concentrations and plotted log (apparent viscosity) vs. log (shear rate) (from 0.01 to 1000 s^−1^) to determine zero-shear viscosities. The Yanang gum flow rate was measured with the range of concentration 0.1 to 2% (*w*/*v*) and over the range of temperature 5–65 °C at 1% concentration. The shear rate can be increased (ahead measurements) or decreased (reverse measures). The shear stress, σ, is a function of the shear rate, γ^•^. The apparent viscosity, η, can be calculated using the equation [11]:σ = ηγ(1)

The power-law model with two parameters was applied to the flow rate of Yanang gum, which is the most commonly applied for engineering applications:σ = k(γ^•^)^n^(2)
where σ is shear stress (Pa), γ^•^ shear rate (s^−1^), and k is the consistency index. The fluid behavior index, n, is different depending on the kind of flow: n < 1 for shear-thinning behavior, n = 1 for Newtonian behavior, and n > 1 for shear-thickening behavior. The parameters k and n were obtained from linear regression analysis.

#### 2.4.3. Dynamic Measurements 

The viscoelastic properties, storage modulus G′, and loss modulus, G″ were investigated using small amplitude oscillatory testing at frequencies ranging from 0.1 to 10 Hz. The strain sweep test was carried out at a constant frequency (0.1 Hz) to estimate the linear viscoelastic area before undertaking any dynamic research [9]. All oscillatory testing was performed at a strain value of 2% within the linear viscoelastic region. A thin film of low viscosity mineral oil was applied to the sample to prevent solvent evaporation during measurements.

### 2.5. Statistical Analysis

All experiments and analytical measurements were run in triplicate. Means of each parameter were analyzed by analysis of variance (ANOVA). Differences between treatments at the 95% (*p* ≤ 0.05) level were considered significant.

## 3. Results and Discussion

### 3.1. Monosaccharide Compositions

The extracted polysaccharides from Yanang leaves consisted of two fractions: crude and purified extracts. The major monosaccharide in the crude and purified extracts was xylose followed by arabinose, galactose, glucose and rhamnose. The xylose content was significantly higher (82%) in the purified extract than the crude extract (76%). The other monosaccharide contents of crude extract were diminished after dialysis, including uronic acid, reduced from 10% in the crude to 7% in the purified extract. Thus, the majority of the polysaccharide gum in Yanang leaves is xylan, which was later confirmed by methylation and spectroscopic analysis.

### 3.2. Glycosidic Linkage Position

Purified Yanang gum was analyzed for linkage patterns (Table 1). Methylation analysis of the purified Yanang extract revealed that it was mainly composed of Terminal-Ara*f*; Terminal-Xyl*p*; 1,3- linked Xyl*p*, and 1,4- linked Xyl*p* with small quantities of branching units: 1,3,5-Ara*f*; 1,2,3- Xyl*p*; 1,2,4-linked Gal*p*; 1,2,3-linked Gal*p*, and 1,3,6-linked Glu*p*. Figure 1a displays the corresponding mass spectra. By comparing mass spectra from the literature with known fragmentation patterns of PMAA, three noteworthy peaks were detected by GC analysis, and they were identified by mass spectroscopic analysis [12].

The terminal pyranosyl pentitol (Terminal Xyl*p*) derivative is a symmetrical 2,3,4-tri-O-methyl alditol acetate, comprising two facile cleavages between adjacent methylated carbons (Figure 1a). Reducing the permethylated sugar with NaBD_4_ introduces an asymmetry that results in a doublet of each major ion (*m*/*z* 101, 102, 117, 118, 161, 162). The combination of the fragmentation patterns suggested that the central chain of Yanang gum comprises 1→3 and 1→4-linked xylosyl residues. Figure 1b shows the symmetrical derivative of 3-linked pentopyranose, which yields nearly equal intensities of *m*/*z* 233 and 234, in addition to *m*/*z* 117 and 118. At the same time, the 4-linked pentopyranose molecules produce almost identical fragments at *m*/*z* 189, 129 and 118 (Figure 1c). The *m*/*z* 118 peak specifies that C-2 must have been methoxylated. The *m*/*z* 129 is produced by loss of acetic acid by β-elimination of *m*/*z* 189. The structure of Yanang gum is a xylan with a backbone chain of 1,3 and 1,4 linkage xylose and side chains of arabinose, galactose and glucose, according to these findings. The structural complexity of xylans from different sources can differ highly. They are, however, mostly made up of a backbone chain of (1,4)-linked -D-xylopyranosyl residues [13]. Regardless of these broad characteristics, the type, number, position and distribution of glycosidic side chains throughout the xylan backbone are all controlled by the source from which the xylan is taken. The backbones of Gramineae (grasses and cereals), Gymnosperms (softwoods), and Angiosperms (hardwoods) are all replaced by -D-glucuronic acid, 4-O-methyl-D-glucuronic acid, and a neutral sugar like -L-arabinose, -D-xylose or -D-galactose [14].

### 3.3. FT-IR Spectroscopy

Figure 2 shows the FT-IR spectra of crude and purified Yanang gum. O-H stretching due to inter-and intra-molecular hydrogen bonding causes the broad absorption band between 3600 and 2500 cm^−1^. The bands at 1730 and 1600 cm^−1^ exhibited a substantial relationship with xylan concentration.

According to the literature, these bands are assigned to distinct vibrations from the carboxylic acid and carboxylate ion group in the 4-O-methyl-D-glucuronic acid substituents in the xylan [15]. The intensity of the band at 1600 cm^−1^ was lower than that of the crude extract, indicating that a carboxylic acid group was eliminated during the dialysis process. In FT-IR, wave numbers between 900 and 1200 cm^−1^ are measured the fingerprint region for carbohydrates. They make it possible to identify important chemical groups in polysaccharides [16]. Xylans with β-(1→4)- and β-(1→3)- linked xylose units in their backbone show a marked difference in their spectral shape due to different conformations [17]. The spectra of xylan with a (1→4)-backbone shows an intense ring and a (COH) side group band at 1047 cm^−1^, whereas the spectrum of (1→3)-linked xylan shows two bands at 1068 and 1026 cm^−1^. Figure 2 shows overlapping bands between 1020–1070 cm^−1^ resulting in a mixed (1→4)- and (1→3)-linked xylose backbone, which was confirmed by methylation analysis. The band occurring at ~1170 cm^−1^ derives from glycosidic bond (COC) vibration of (1→4)- and (1→3)-linked xylose [17].

### 3.4. NMR Spectroscopy

The ^1^H-NMR and ^13^C-NMR analyses were performed in order to learn more about Yanang gum’s anomeric linkage arrangement (Figure 3 and Table 2). The ^13^C-NMR spectrum of Yanang gum (Figure 3a shows the presence of a mixed linkage of β-(1→3) and β-(1→4)-D-xylan. Five strong signals at 101.5, 75.5, 72.6, 69.1 and 65.1 ppm distinguish the major 1,4-linked-D-Xyl*p* units which assigned to C-1, C-4, C-3, C-2 and C-5 of the β-D-Xyl*p*, respectively. The 1,3-linkage is represented by 98.9, 75.5, 73.3, 70.4 and 67.0 ppm, and corresponds to C-1, C-4, C-3, C-2 and C-5 of β-D-Xyl*p* [18,19]. The signals were assigned at 108.9, 83.9, 81.2, 77.6 and 61.3 ppm to C-1, C-4, C-2, C-3 and C-5 of the α-L-Araf residues, respectively. The carboxylic group from C-6 of the glucuronic acid residue in the xylan causes the resonance at 170.15 [20]. Signals at 52.76 and 29.45 ppm are consistent with methyl and acetyl groups of glucuronic acid residue [21]. Figure 3b shows the proton NMR spectrum of Yanang gum. The spectrum shows the presence of mixed-linkage β-(1→3)/β-(1→4)-D-xylans. The 1,4-linkage represents about 49.6% of the total linkages in Yanang gum, whereas the 1,3-linkage reveals about 21.8% (Table 1). The anomeric proton was attributed at 4.44 ppm to the β-(1→4)-linked xylose and the doublet at δ 4.5–4.6 to the anomeric signal of the 1→3-linked xylose [18]. Figure 3 shows the pre-saturation COSY spectra of Yanang gum. Figure 3c,d is an enlargement for β-(1→4)- and β-(1→3)- linked xylose, respectively. The connectivity of a monosaccharide ring from the COSY spectrum was established. Table 2 summarizes the results of a partial assignment of the COSY spectrum.

### 3.5. Molecular Characterization

High performance size-exclusion chromatography was used to explore the molecular characterization of purified Yanang gum (HPSEC). Yanang gum had a molecular weight of 3819 kDa on average, significantly higher than various literature values reported for xylan. Oat spelt xylan is reported to have molecular masses within the range of 50–200 kDa [22]. The published values for rye arabinoxylans are 519–770 kDa, and for wheat, 255 kDa [23]. Moreover, Xylan- xyloglucan complexes in the cell walls of olive pulp and common bean (*Phaseolus vulgaris* L.) have molecular weights of 100–2000 kDa and 1576–3505 kDa, respectively [24,25]. Acidic xylans with molecular weights ranging from 1300 to1700 kDa exude from the leaves of *Phormium tenax* and *Phormium cookianum* [26]. The quantification of the molecular mass depends on the source of extraction, their estimation method and the sample preparation. The polydispersity index of 1.7 specified a broad molecular distribution of Yanang gum. The HPSEC method provided the radius of gyration (Rg) as 120.4 nm and the intrinsic viscosity [η] as 14.57 dL/g. The significant differences in the molecular masses result from polymer conformations or chain aggregation, which may arise from different sample solubilization methods or operating conditions.

### 3.6. Rheological Properties

#### 3.6.1. Steady Shear Measurements

Figure 4 shows typical viscosity curves in the range of 10^−2^–10^3^ s^−1^ of the shear rate, as a function of Yanang gum concentrations. The shear rate had no influence on the viscosity at a lower concentration (0.1%). Shear-thinning behavior becomes more specious at higher concentration events and at lower shear rates, and this becomes more noticeable as the polymer concentration increases.

Figure 5a,b show the flow curves of Yanang gum at different concentrations, measured in forward and backward directions, respectively. In general, the Yanang gum solutions display shear-thinning behaviors, which could be explained that as the rate of deformation increased, their viscosities decreased. When this occurs, the shear-thinning behavior could be fit to the power-law model [11]. Figure 5 depicts the fits of the experimental results to Equation (2) which correspond precisely to the experimental data. The values of the parameters k and n and the determination coefficient, R^2^, are also given in Table 3.

According to the power-law model, Newtonian behavior at 0.1 percent Yanang gum concentration exhibits a straight-line relationship between shear stress and shear rate, including the fluid behavior index (n), which is near 1. The results demonstrate that the shear stress–shear rate relationship is non-linear, implying that at large concentrations, Yanang gum acts as a non-Newtonian fluid. As the gum concentration increases, k increases and n decreases. If k increases in concentration, this indicates that the viscosity of the gum increased with higher concentration. Moreover, the fact that n of concentration more than 0.1% (*w*/*v*) are less than unity indicates that Yanang gum displays shear-thinning behavior. The decrease in n, with increasing concentration, implies a more important shear-thinning behavior of the system. This behavior is similar to other polymer systems such as tehineh [27], fenugreek paste [7] and mucilage (*Opuntia ficus indica*) gum [28].

Figure 6 shows the flow curves of Yanang gum (1% *w*/*v*) at different temperatures (5–65 °C), both in the forward (increasing shear rate) and backward (decreasing shear rate) directions. The forward curves show higher shear stress than those of the backward curves at high temperatures. At low temperatures, they also display a similar curve in both forward and backward readings. The existence of hysteresis loops indicates that the molecular structure of Yanang gum has been sheared, resulting in a drop in apparent viscosity. The shear stress–shear rate relationship is non-linear as shown in Table 4, demonstrating that Yanang gum is a non-Newtonian fluid. Moreover, the fact that n is less than unity indicates that Yanang gum displays pseudo-plastic (shear-thinning) behavior. This is due to the swelling and solubility of Yanang gum at higher temperatures, as well as the disruption of the gel network caused by increased shear strain.

For both forward and backward measurements, the temperature dependency of the consistency coefficient, k, is shown in Figure 7. From research work by Abu-Jdayil et al. [27] and Medina-Torres et al. [28], the impact of temperature on k in the forward direction is given by:k = exp (2.9543 − 0.061T)(3)
where T is the temperature in degrees Celsius, and the effect is given by:k = exp (3.0265 − 0.0764T)(4)

The consistency coefficient, k, clearly decreases with increasing temperature. Alternatively, Figure 7 shows how temperature affects the flow behavior index, n. The influence of temperature on n in the forward measurement is given by:n = 0.1861 + 0.015T(5)
where T is the temperature in degrees Celsius, and in the backward measurement the effect is given by:n = 0.1873 + 0.0111T(6)

Unlike the consistency coefficient, k, n increases little as temperature rises. Yanang gum exhibits flow behavior similar to that of a pseudo-plastic food fluid (n < 1). Tehineh showed that the consistency coefficient, k, and the flow behavior index, n, both of which are connected to Yanang gum and *Opuntica ficus indica* gum, fluctuate with temperature [27,28].

#### 3.6.2. Dynamic Measurements

Yanang gum’s viscoelastic behavior was studied at temperature 25 °C throughout a frequency range of 0.1 to 10 Hz. Figure 8 shows the master curve of storage modulus, G′, and loss modulus, G″ in relation to frequency. The mechanical spectra of Yanang gum revealed a diluted solution with G″ greater than G′ over the frequency range, but they approached each other at higher frequencies. When the frequency increased, the viscoelastic fluids, the storage modulus, G′ increased faster than the loss modulus, G″. With increased concentration, the cross-over changed to a lower frequency, indicating a stronger elastic component. The storage modulus, G′, and loss modulus, G″ curves cross at the center of the frequency range in a concentrated solution (1% *w*/*v*), indicating a clear propensity for more solid-like behavior at higher frequencies. At the higher concentrations (2% and 3% *w*/*v*) the storage modulus was studied, G′ showed a weak dependence on frequency and was higher than loss modulus, G″, with G′ and G″ converging at a lower frequency. This behavior is typical of a weak gel system which corresponds to a system of intermolecular entanglements [29,30]. The time sweep results are presented in Figure 9 as storage modulus, G′, vs. time in order to determine the gelling properties of Yanang gum. G′ increased rapidly for the first 20 min then gradually increased slowly until it reached an apparent plateau. It is noteworthy that the former polymer remained stable at 25, 30 and 40 min at 5, 15 and 25 °C, respectively. Although Yanang gum had similar mechanical qualities, its gel strength was higher at low temperatures.

Figure 10 reveals that the frequency dependence of complex viscosity, η*, and the shear rate dependence of apparent viscosity, η, were found to be nearly identical at varying concentrations of Yanang gum solutions. The complex viscosity, η*, was slightly lower than the apparent viscosity, η, for each solution and can be attributed to the heterogeneity of polysaccharide gum that undergoes aggregation and also to its highly branched structure. This phenomenon was also seen in other polysaccharide dispersions such as *Aeromonas* gum [31] and exopolysaccharides from the *Escherichia coli* strain S61 [32]. The deviation from the Coz–Merz rule for Yanang gum can be attributed to the formation of aggregates with an increase in concentration, complex viscosity, η* lower than apparent viscosity and occurring at a lower shear rate or frequency, suggesting that the formation of aggregates at high concentrations is easier than at low concentrations.

## 4. Conclusions

It is concluded that the polysaccharide gum from Yanang leaves (*Tiliacora triandra*) is xylan mainly consisting of xylose with small amounts of other neutral sugars and uronic acid. The main structure of the polysaccharide gum from Yanang leaves is mixed β-(1→3)- and β-(1→4)-D-xylans. Yanang gum is a semi-solid substance that exhibited non-Newtonian behavior. The dynamic measurements were relative to concentrated solutions of Yanang gum. The steady and oscillatory shear viscosities are found to be similar at comparable rates. From this study, Yanang gum may be as useful as other natural gums for food industrial applications

## Figures and Tables

**Figure 1 foods-11-02003-f001:**
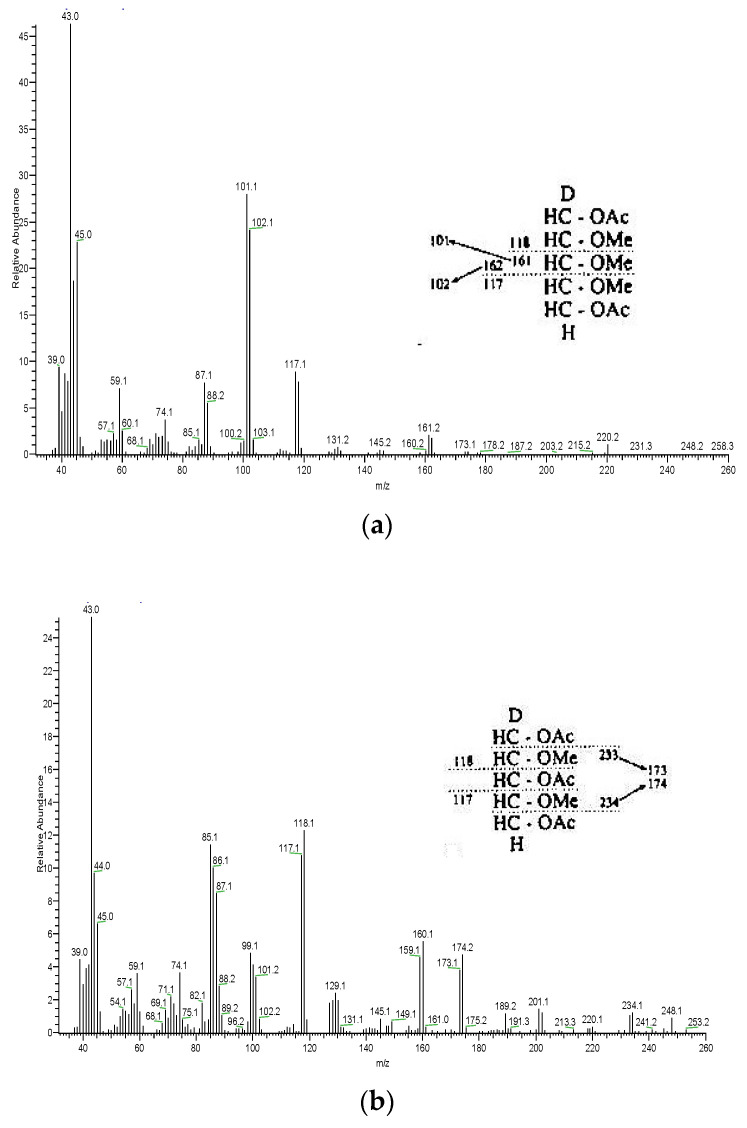
Mass spectrum of purified Yanang gum. (**a**) Terminal Xyl*p* (**b**) 1,3-Xyl*p* (**c**) 1,4-Xyl*p*.

**Figure 2 foods-11-02003-f002:**
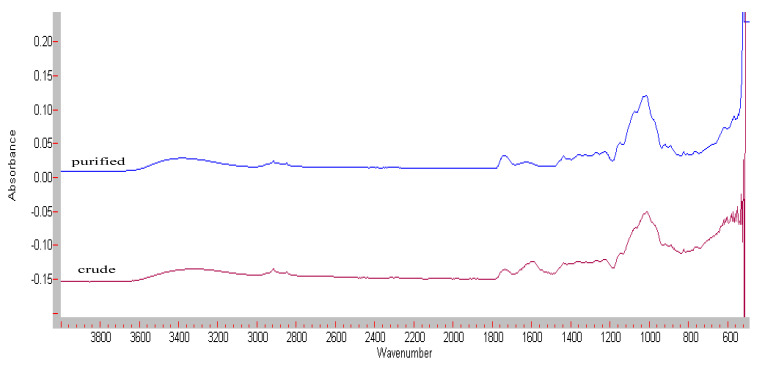
FT-IR spectra of crude and purified Yanang gum.

**Figure 3 foods-11-02003-f003:**
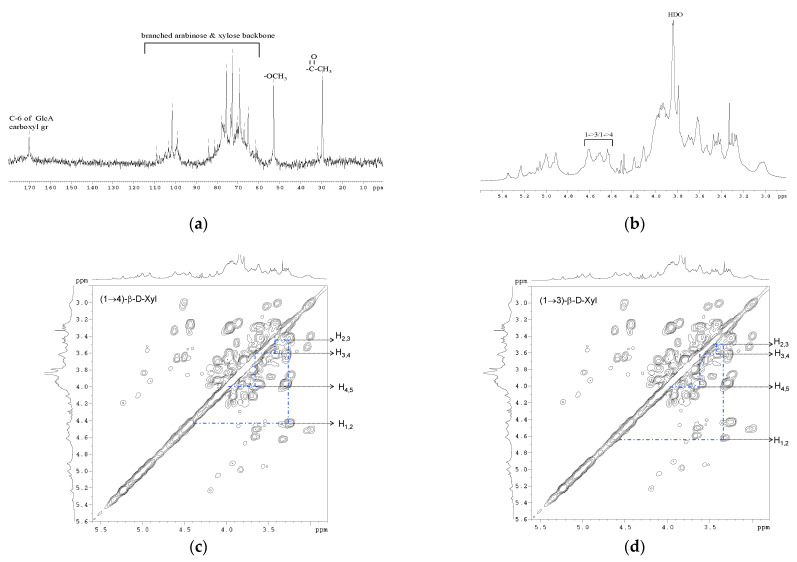
NMR spectra and COSY spectrum of purified Yanang gum. (**a**) ^13^C-NMR spectra, (**b**) ^1^H-NMR spectra. (**c**) COSY spectrum of Yanang gum, Dashed lines is the correlation of protons of 4-β-D-xylose. (**d**) COSY spectrum of Yanang gum, Dashed lines is the correlation of protons of 3-β-D-xylose.

**Figure 4 foods-11-02003-f004:**
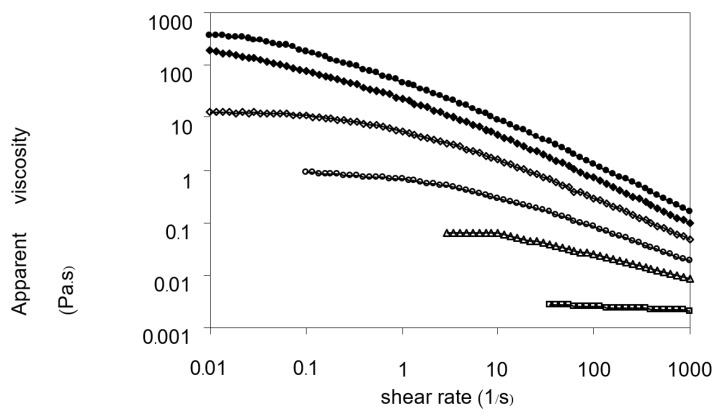
Effect of Yanang gum concentration on the steady-shear viscosity at 25 °C. (☐) 0.10%, (Δ) 0.25%, (ο) 0.50%, (◊) 1.00%, (♦) 1.50%, (•) 2.00%.

**Figure 5 foods-11-02003-f005:**
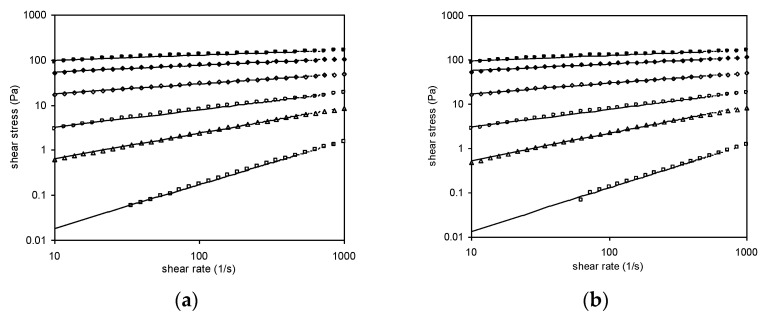
Flow curves of Yanang gum at different concentrations by increasing and decreasing the shear rate. (☐) 0.10%, (Δ) 0.25%, (ο) 0.50%, (◊) 1.00%, (♦) 1.50%, (•) 2.00%. (**a**) Forward measurements (increasing the shear rate); (**b**) Backward measurements (decreasing the shear rate).

**Figure 6 foods-11-02003-f006:**
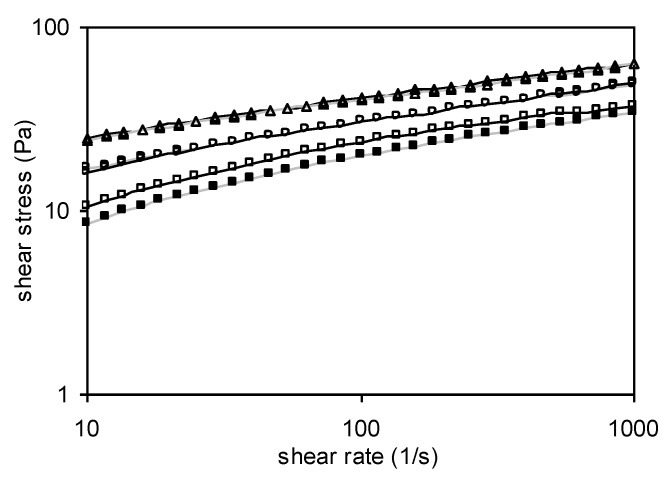
Hysteresis loops of the flow curves of Yanang gum at different temperatures. (open symbol = forward measurements and filled symbol = backward measurement); (Δ) 5 °C, (ο) 25 °C, (☐) 65 °C.

**Figure 7 foods-11-02003-f007:**
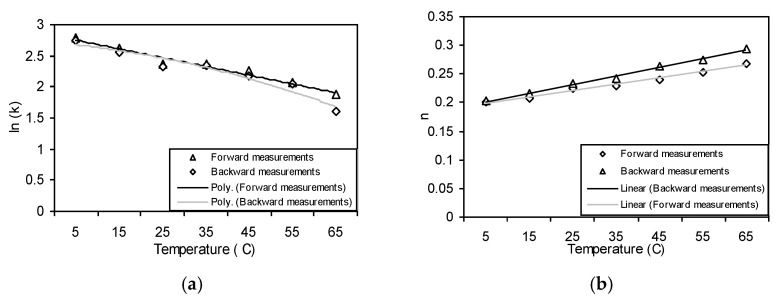
Dependence of the consistency coefficient and flow behavior index of Yanang gum on the temperature. (**a**) the consistency coefficient (**b**) the flow behavior index.

**Figure 8 foods-11-02003-f008:**
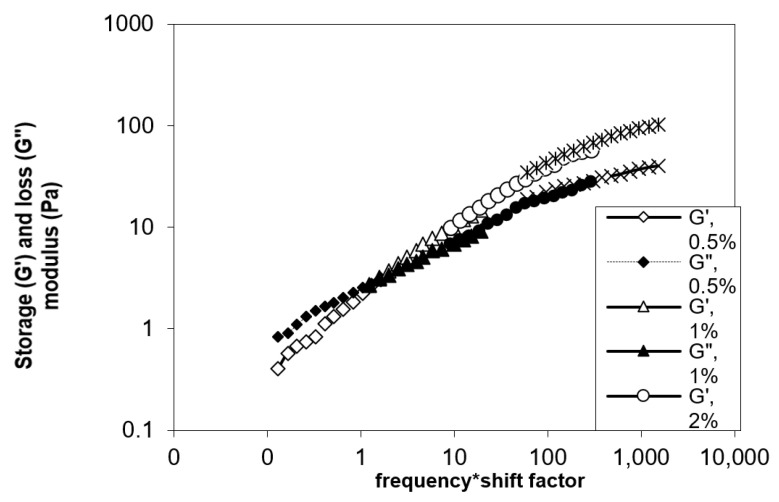
Master curve of storage (G′) and loss (G″) modulus as a function of frequency for Yanang gum at different concentrations.

**Figure 9 foods-11-02003-f009:**
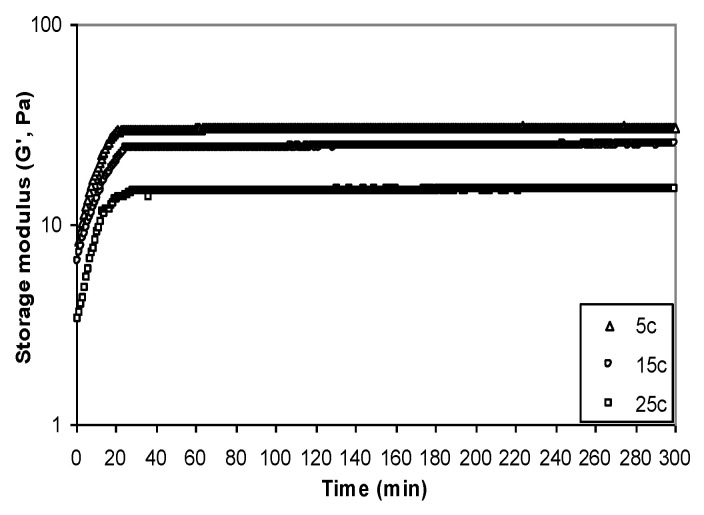
Evolution of storage modulus (G′) with time of Yanang gum 2% (*w*/*v*).

**Figure 10 foods-11-02003-f010:**
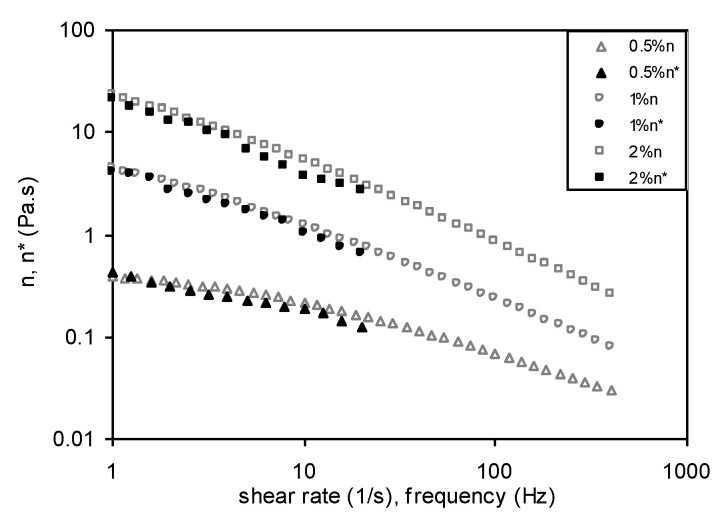
Cox-Merz plot of steady shear (closed symbol) and dynamic (open symbol) viscosities for 0.5, 1.0 and 2.0% Yanang gum.

**Table 1 foods-11-02003-t001:** Neutral sugar linkage composition (relative abundance %) of purified Yanang gum.

Chemical Name	Deduced Linkage	Relative Abundance (%) *
1,4-di-O-acetyl-(1-deuterio)-2,3,5-tri-O-methyl pentitol	Terminal-Ara*f*	5.755 ± 0.18
1,3,4,5-tetra-O-acetyl-(1-deuterio)-2-O-methyl pentitol	1,3,5-Ara*f*	0.504 ± 0.08
1,5-di-O-acetyl-(1-deuterio)-2,3,4-tri-O-methyl pentitol	Terminal Xyl*p*	14.553 ± 0.54
1,3,5-tri-O-acetyl-(1-deuterio)-2,4-di-O-methyl pentitol	1,3-Xyl*p*	21.884 ± 0.64
1,4,5-tri-O-acetyl-(1-deuterio)-2,3-di-O-methyl pentitol	1,4-Xyl*p*	49.637 ± 0.71
1,2,3,5-tetra-O-acetyl-(1-deuterio)-4-O-methyl pentitol	1,2,3-Xyl*p*	0.549 ± 0.08
1,2,4,5-tetra-O-acetyl-(1-deuterio)-3,6-di-O-methyl hexitol	1,2,4-Gal*p*	1.687 ± 0.09
1,2,3,5-tetra-O-acetyl-(1-deuterio)-4,6-di-O-methyl hexitol	1,2,3-Gal*p*	3.767 ± 0.12
1,3,5,6-tetra-O-acetyl-(1-deuterio)-2,4-di-O-methyl hexitol	1,3,6-Glu*p*	1.663 ± 0.07

* Relative abundance (%) calculated from the ratio of peak area.

**Table 2 foods-11-02003-t002:** Assignment of ^1^H NMR spectrum of Yanang gum.

Type of Linkage	Chemical Shifts (ppm)				
	H-1	H-2	H-3	H-4	H-5
1,3-Xyl*p*	4.64	3.36	3.49	3.65	3.98/3.77
1,4-Xyl*p*	4.46	3.29	3.46	3.57	3.98/3.65

**Table 3 foods-11-02003-t003:** Rheological parameter of the power law model of Yanang gum at different concentrations.

Concentration (%)	Forward Measurements			Backward Measurements		
	k (Pa·s^n^)	n	R^2^	k (Pa·s^n^)	n	R^2^
0.10	0.0020	0.9711	0.9995	0.0014	0.9817	0.9968
0.25	0.1716	0.5716	0.9991	0.1297	0.6140	0.9962
0.50	1.3133	0.3889	0.9957	1.2028	0.4004	0.9959
1.00	10.5880	0.2248	0.9960	10.1860	0.2330	0.9952
1.50	39.71	0.1452	0.9903	39.308	0.1520	0.9889
2.00	75.581	0.1143	0.9743	72.474	0.1193	0.9721

R^2^ is the coefficient of determination, k is the consistency coefficient, n is the flow behavior index.

**Table 4 foods-11-02003-t004:** Rheological parameter of the power low model of Yanang gum (1% *w*/*v*) at different temperatures.

Temperature (°C)	Forward Measurements			Backward Measurements		
	k (Pa·s^n^)	n	R^2^	k (Pa·s^n^)	n	R^2^
5	16.231	0.2002	0.9981	15.548	0.2021	0.9983
15	13.712	0.2064	0.9977	12.949	0.2156	0.9971
25	10.588	0.2248	0.9960	10.186	0.2330	0.9952
35	10.545	0.2294	0.9889	10.543	0.2419	0.9930
45	9.6783	0.2397	0.9777	8.9007	0.2632	0.9877
55	7.8844	0.2529	0.9886	7.7036	0.2751	0.9895
65	6.4982	0.2675	0.9785	4.9293	0.2928	0.9854

R^2^ is the coefficient of determination, k is the consistency coefficient, n is the flow behavior index.

## Data Availability

Data is contained within the article.

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
