# Peer review of "Structural and Rheological Properties of Yanang Gum (Tiliacora triandra)"

_foods, 2022, doi:10.3390/foods11142003_

Round 1

Reviewer 1 Report

Abstract

The text must be in the third person

The introduction part in the abstract is lost

Specify the type of fluid obtained.

Add a conclusión

Introduction

It is too short . Must be improved.

The redaction must be improved

Only 3 references for an introduction?

Material san method

L 45.Describe the method

 A section of polysaccharide extraction must be added

L104. Added the condition of method

Improve redaction

L105. We made plots…. Eliminate

L120. Added the condition of strain and frequency sweep

Added the references of methods employee

Result and discussion

All equations must be improved (especially the shear rate symbol)

A low number of references

Improve the quality of Figure 1

Figure 2. It's difficult to analyze the Figures and the symbol for each sample

L259. Fix the citation format

Table 3b, specify the concentration of solutions

Figure 3. The G’ and G’’ of gums at different concentration present the same behavior? Explain ¿How can variate the complex viscosity?

Author Response

Dear reviewer,

Please see the detailed below for response to reviewer;

1. Abstract: The text must be in the third person. The introduction part in the abstract is lost. Specify the type of fluid obtained.

Response: Thank you for your suggestion, we had re-written the abstract.

2. Introduction: It is too short . Must be improved. The redaction must be improved Only 3 references for an introduction?

Response: Thank you for your suggestion, we had already added literature review in introduction part.

3. Material san method: L 45.Describe the method, A section of polysaccharide extraction must be added, L104. Added the condition of method, Improve redaction, L105. We made plots…. Eliminate,L120. Added the condition of strain and frequency sweep, Added the references of methods employee

Response: Thank you for your suggestion, we had already corrected and added information following reviewer suggestion.

3. Result and discussion: All equations must be improved (especially the shear rate symbol), A low number of references, Improve the quality of Figure 1

Figure 2. It's difficult to analyze the Figures and the symbol for each sample

L259. Fix the citation format, Table 3b, specify the concentration of solutions, Figure 3. The G’ and G’’ of gums at different concentration present the same behavior? Explain ¿How can variate the complex viscosity?

Response: Thank you for your suggestion, we had already corrected and added information following reviewer suggestion.

Best regards,

Ratchadaporn Oonsivilai

Reviewer 2 Report

The undertaken study is interesting and with elements of novelty; however, the research should be more justified, i.e. introduction section should be extended in this regard. More specific comments are provided below.

Lines 14 and 15: Standardize the tense of verbs.

Line 42: Remove “then”

Line 55: I recommend replacing the “prepared with” with “composed of”.

Section 2. There is no information on test repetitions and statistical analysis of the experimental data.

Section 3.1: Please provide data with statistical analysis (in Table) for the monosaccharide compositions.

Please justify the rheological measurements at various temperatures. The concentration range should also by justified.

The graphs in Figure 1 are too small.

The graphs in Figure 2 are too small. There are no symbols in brackets provided in Fig. 2A-C captions.

Figure 2B: Why the forward measurements and backward measurements are not depicted in one figure? It would enable one to conclude about hysteresis loop.

Line 268: What does “with more viscous” mean?

Line 289: Please explain the statement “disruption of the gum”.

Equations 3 and 4 as well as 5 and 6: How were these equations determined, by what software? When substituting the values of temperatures in these equations, the model values of the k coefficient are quite different from the experimental ones. The same concerns the model n values. Please verify the correctness of these equations.

Fig. 3B and 3C: Explain the symbols used in the legends.

Lines 356-360: This is a repetition of the results. Convert into real conclusion.

Lines 361-364: The sentence is unintelligible.

Lines 365-366: Too general statement and not resulting from the conducted research.

Author Response

Dear Reviewer,

Please see the detailed below for the response;

1. Lines 14 and 15: Standardize the tense of verbs. Line 42: Remove “then”

Line 55: I recommend replacing the “prepared with” with “composed of”.

Response: Thank your for your comments, we had corrected the text following your suggestion.

2. Section 2. There is no information on test repetitions and statistical analysis of the experimental data.

Response: Thank your for your comments, we had corrected the text following your suggestion.

3. Section 3.1: Please provide data with statistical analysis (in Table) for the monosaccharide compositions. Please justify the rheological measurements at various temperatures. The concentration range should also by justified.

Response: Thank your for your comments, we had corrected the text following your suggestion.

4. The graphs in Figure 1 are too small. The graphs in Figure 2 are too small. There are no symbols in brackets provided in Fig. 2A-C captions. Figure 2B: Why the forward measurements and backward measurements are not depicted in one figure? It would enable one to conclude about hysteresis loop. Fig. 3B and 3C: Explain the symbols used in the legends. Lines 356-360: This is a repetition of the results. Convert into real conclusion. Lines 361-364: The sentence is unintelligible. Lines 365-366: Too general statement and not resulting from the conducted research. 

Response: Thank your for your comments, we had corrected the figure following your suggestion.

5. Line 268: What does “with more viscous” mean? Line 289: Please explain the statement “disruption of the gum”. Equations 3 and 4 as well as 5 and 6: How were these equations determined, by what software? When substituting the values of temperatures in these equations, the model values of the k coefficient are quite different from the experimental ones. The same concerns the model n values. Please verify the correctness of these equations.

Response: Thank your for your comments, we had corrected the text, added more detail following your suggestion.

Best regards,

Ratchadaporn oonsivilai

Reviewer 3 Report

The manuscript submitted for evaluation covers the structural and rheological properties of Yanang gum. The authors analyzed the saccharide composition as well as the structure using instrumental methods. The work methodology is well described. You can have reservations about the description of the research material - was the material homogeneous, how was it selected? Was it accidental material? Another suggestion - please write down how many repetitions the determinations have been made. Were the extracts prepared in several or one repetition?

In my opinion, the Introduction does not contain information on the research material. Most of the Introduction is part of the discussion.

I suggest preparing / editing this chapter

I also suggest adding more discussion

Since the authors in the conclusion refer to the possible use of Yanang gums in the food industry, it is worth mentioning the use of gums. Please correct the References - Article Title and Volume – Italic.

Author Response

Dear Reviewer,

Please see the detail below for the response;

1.You can have reservations about the description of the research material - was the material homogeneous, how was it selected? Was it accidental material? Response : Thank you for your suggestion, we already explained in revision manuscript.

2.Another suggestion - please write down how many repetitions the determinations have been made. Were the extracts prepared in several or one repetition?

Response: Thank you for your suggestion, we already explained in revision manuscript.

3.In my opinion, the Introduction does not contain information on the research material. Most of the Introduction is part of the discussion. I suggest preparing / editing this chapter

Response: Thank you for your suggestion, we already explained and added literature review in introduction part in revision manuscript.

4. I also suggest adding more discussion, Since the authors in the conclusion refer to the possible use of Yanang gums in the food industry, it is worth mentioning the use of gums. Please correct the References - Article Title and Volume – Italic.

Response: Thank you for your suggestion, we already added information in discussion and conclusion parts in revision manuscript.

Best regards,

Ratchadaporn Oonsivilai

Reviewer 4 Report

Review foods-1764268

For each sample, did the authors collect multiple spectra, triplicate of spectra, for instance? In addition, whether the scan was smoothened, baseline corrected, normalised for qualitative interpretation of spectra. 

What internal standard was applied for the NMR analysis? In addition, for Table 3, RMSE or SSE should be added as R^2 is not sufficient for validating a model.  For instance, Food Research International, 144, 110355.

It is suggested to add a section ‘statistical analysis’ after section 2.4.4. 

Figure 2A: better to use ‘Apparent viscosity’ rather than ‘Viscosity’ for the y-axis as they are non-Newtonian fluids. 

The discussion should be enhanced by comparing with other polysaccharide-based gums. For instance, International Journal of Biological Macromolecules, 117, 294-300.

Author Response

Dear Reviewer,

Please see the detailed below for response;

1.For each sample, did the authors collect multiple spectra, triplicate of spectra, for instance? In addition, whether the scan was smoothened, baseline corrected, normalised for qualitative interpretation of spectra. 

What internal standard was applied for the NMR analysis? In addition, for Table 3, RMSE or SSE should be added as R^2 is not sufficient for validating a model.  For instance, Food Research International, 144, 110355.

Response: Thank you for your comment, we followed the method of referenced listed in revised manuscript. 

2. It is suggested to add a section ‘statistical analysis’ after section 2.4.4. 

Response: Thank you for your comment, we already added statistical analysis part after section 2.4.4.

3. Figure 2A: better to use ‘Apparent viscosity’ rather than ‘Viscosity’ for the y-axis as they are non-Newtonian fluids. 

Response: Thank you for your comment, we already corrected the text following your suggestion.

4.The discussion should be enhanced by comparing with other polysaccharide-based gums. For instance, International Journal of Biological Macromolecules, 117, 294-300.

Response: Thank you for your comment, we already compared with other polysaccharide-based gums in revised manuscript.

Best regards,

Ratchadaporn oonsivilai